# Effects of Combined Vibration Ergometry and Botulinum Toxin on Gait Improvement in Asymmetric Lower Limb Spasticity: A Pilot Study

**DOI:** 10.3390/jfmk10010041

**Published:** 2025-01-21

**Authors:** Harald Hefter, Dietmar Rosenthal, Sara Samadzadeh

**Affiliations:** 1Department of Neurology, University of Düsseldorf, Moorenstrasse 5, 40225 Düsseldorf, Germany; dietmar.rosenthal@med.uni-duesseldorf.de (D.R.); sara.samadzadeh@yahoo.com (S.S.); 2Charité–Universitätsmedizin Berlin, Corporate Member of Freie Universität Berlin and Humboldt-Unverstät zu Berlin, Experimental and Clinical Research Center, 13125 Berlin, Germany; 3Department of Regional Health Research and Molecular Medicine, University of Southern Denmark, 5230 Odense, Denmark; 4Department of Neurology, Slagelse Hospital, 4200 Slagelse, Denmark

**Keywords:** vibration ergometry, botulinum neurotoxin type A, botulinum toxin therapy, lower limb spasticity, functional improvement of gait, muscle spindle activity

## Abstract

**Objective:** Botulinum neurotoxin type A (BoNT/A) injections and the new vibration ergometry training (VET) are studied for their combined effect on improving functional mobility in patients with asymmetric lower limb spasticity. **Method:** Gait was analyzed using the Infotronic^®^ system, which measures ground reaction forces and foot contact patterns by means of special force-sensitive shoes strapped over feet or street shoes. Gait was measured several times, depending on the protocol patients underwent. Seven patients with asymmetric lower limb spasticity were analyzed according to the control protocol (CG-group): after a baseline walk of 20 m (NV-W1) patients received their routine BoNT/A injection and had to walk the same distance a second time (NV-W2). Approximately 3–5 weeks later, they had to walk a third time (NV-W3). A further seven patients (VG-group) were analyzed according to the vibration protocol: after a baseline walk (V-W1), patients underwent a first vibration training (VET1), walked a second time (V-W2), received their routine BoNT/A injection, and walked a third time (V-W3). About four weeks later, they had to walk again (V-W4), received another vibration training (VET3), and walked a fifth time (V-W5). At least six months after the analysis according to the vibration protocol, these patients were also analyzed according to the control protocol. Eleven gait parameters were compared between the CG- and VG-group, and within the VG-group. **Result:** Patients in the VG-group experienced a significant improvement in gait four weeks after BoNT/A injection, unlike the patients in the CG-group. VG-patients also showed improved gait after two VET sessions. However, there was no further functional improvement of gait when BoNT/A injections and VET sessions were combined. **Conclusions:** BoNT/A injections enhance functional mobility in patients with mild asymmetric leg spasticity. VET also induces an immediate gait improvement and offers a further treatment approach for leg spasticity. Whether combining BoNT treatment and vibration training offers superior outcomes compared to either treatment alone requires further investigation.

## 1. Introduction

Intramuscular injections of botulinum neurotoxin type A (BoNT/A) reduce activity of extra- and intrafusal muscle fibers [1,2]. This is the basis for why BoNT/A injections effectively reduce elevated muscle tone and spasticity [3]. Because of its efficacy, BoNT/A treatment has received level A recommendations from the American Academy of Neurology and the British Royal College of Physicians for the treatment of spasticity [4,5]. Whether BoNT/A treatment of lower leg spasticity also leads to a functional improvement remains a matter of debate [6].

During long-term BoNT/A treatment of spasticity, the development of neutralizing antibodies is a frequent dose-dependent side-effect [7]. Therefore, techniques and training methods that reduce spasticity, support and enhance BoNT-injection therapy, and thereby allow for dose reduction are highly valued.

It is well known that vibration of tendons and muscles significantly alters the tonic vibration reflex and reduces muscle spindle activity [8,9,10,11]. In clinical practice, whole-body vibration (WBV) or segmental limb vibration (SLV) have successfully been used to influence spasticity in patients with spinal cord injury (SCI; [12,13]), in patients with infantile cerebral palsy (ICP; [14,15,16,17,18,19]), in hemiparetic post-stroke patients [20,21,22,23], or for multiple sclerosis (MS) ([24,25], for reviews see also [26,27]).

But little is known about the combined application of vibration and BoNT. In MS patients, segmental leg muscle vibration has been combined with BoNT injections. An enhancement of BoNT efficacy could not be detected. This study, however, used the fatigue severity scale as the primary outcome measure, rather than spasticity or gait [25].

Vibration does not only influence muscle spindles but also has a widespread effect on cellular metabolism, blood supply, tissue growth, and bone turn-over [28,29,30,31]. All systems in the body, especially the cardiovascular system, react to vibration [32,33]. During vibration, cross sections of the large as well as the small vessels deform and the total peripheral resistance (TPR) increases [33]. A special safety concern with vibration is resonance, which should be avoided to prevent damage to inner organs, the brain, and vessels [33]. In athletes, who use vibration to increase muscle strength and endurance, some experiments have been performed to control the effect of vibration not only on cardiovascular parameters, such as cardiac output, blood pressure, and TPR, but also on markers, such as S100B, indicating brain damage [34]. The recommendations for vibration training include: (1) high transmission factor to the head and the resonance frequency range (below 20 Hz) should always be avoided, (2) low amplitudes (1–2 mm) should be used in vibration training for leisure sport, (3) the exposure duration for each vibration training should be very short (20–60 s), (4) vibration training should be avoided for people with coronary disease or hypertension [33]. Notably, no studies have yet assessed the risk of inducing cardiovascular events during or after WBV.

We, therefore, were pleased to have access to vibration ergometry training (VET), which was originally invented for the training of bicycle racing professionals [35,36]. During this special type of vibration technique, subjects or patients sit on a bike with the crank completely decoupled from the bottom bracket and pedals which can be vibrated with an eccentric motor at different amplitudes and frequencies (for details see methods). In contrast to WBV, this special type of vibration does not cause vibration of the trunk and head [37] and avoids an increase of S100B [34] and resonance phenomena in the body, except the legs.

Since VET appears to be much safer for the treatment of spasticity in patients with heart and vessel diseases than WBV, it was used in the present study. The hypothesis was that the application of VET and BoNT/A injections effectively reduce spasticity and enhance each other. To evaluate whether a functional improvement in the lower legs can be achieved through the combined application of VET and BoNT/A injections, several gait parameters were used as outcome measures.

## 2. Materials and Methods

The present study (BONTVIBR) was registered under the number 2017064344 with the local ethics committee of the University of Düsseldorf (Germany). Before the VET prototype could be used for clinical applications, the entire setup of the VET laboratory had to be approved by the Department of Technical and Safety Affairs of the University of Düsseldorf (Germany). Thereafter, the local ethics committee did not have objections to the use of the vibration ergometer as part of our general program to optimize BoNT therapy and reduce antibody formation.

### 2.1. Patients

Patients were recruited from the botulinum toxin outpatient department of the University Hospital Düsseldorf (Germany). All patients were treated on a regular basis, every 3 months. Neither treatment cycle duration, nor dose, nor injection scheme was altered due to this study.

Inclusion criteria were: (1) age: 18–80 years, (2) stable asymmetric lower limb spasticity for at least two years, (3) indication for botulinum toxin therapy of the more affected leg only, (4) ongoing BoNT/A therapy for at least 6 months, (5) patient’s confirmation that he was able to sit on a bicycle (6) ability to walk, alone, a distance of at least 50 m with or without a cane or an ankle-foot orthosis (AFO) or orthopedic shoes, (7) informed consent.

Exclusion criteria were: (1) botulinum toxin treatment during the last 3 months, (2) known stenosis or aneurysm of the aorta or leg arteries, (3) patients under legal care.

Fourteen patients were recruited and stratified according to pre-test VET of 5 minutes’ duration (see Figure 1). Seven patients succeeded in performing the test VET (VG group) and were analyzed according to the vibration protocol. At least 6 months later, these patients were also analyzed according to the control protocol (see Figure 1, left side).

Seven patients did not manage to complete the pre-test VET (CG group) and were analyzed only according to the control protocol as a control group (see Figure 1, right side). There were two reasons for the inability to complete the pre-test VET: (i) in one patient, their legs were too short for the device, (ii) in six patients (who managed to perform cyclic leg movements with a load of 20 Watt [34] as inclusion criterion 6 indicated that leg muscles were only mildly paretic (MRC^®^-class 4 to 5)), the onset of vibration enhanced leg spasticity to such an extent that the cyclic leg movements could not be continued for 5 min.

Demographical and treatment-related data of the VG patients and of the CG patients (Table 1) are presented in Section 3.1.

### 2.2. Design of the Study (Figure 1)

In Figure 1, the design of the study is summarized. Since none of the patients had had experience with vibration ergometry, a pre-test was performed (see Figure 1). Those seven patients who managed to complete the pre-test were collected in the vibration group (VG group). The other 7 patients, who did not complete the pre-test, were collected in the control group (CG group; see Figure 1, upper part).

The seven VG patients were first analyzed according to the vibration protocol (Figure 1, left side) and at least 6 months later, also according to the control protocol (Figure 1, middle part). The CG patients were only analyzed according to the control protocol (Figure 1, right side).

According to the vibration protocol (Figure 1, left side), patients had to walk a first time (V-W1) at the baseline visit, then were vibrated a first time (VET1), and had to walk a second time (V-W2). They were injected with BoNT/A and underwent VET a second time (VET2), then they had to walk a third time (V-W3) at the baseline visit. Four weeks later, these patients had to walk again (V-W4), receive a further third VET (VET3), and had to walk a fifth time (V-W5).

According to the control protocol (Figure 1, right side), patients were not vibrated, had to walk a first time (NV-W1), were injected with BoNT/A, and had to walk a second time (NV-W2). Four weeks later, these patients had to walk a third time (NV-W3).

### 2.3. Details of Vibration Ergometry Training (VET)

For the present investigation, a prototype of a vibration ergometer was available. This device had been developed by Dieter Quarz [35,36] in collaboration with the German Sport University Cologne (Germany). So far, no recommendations for VET in patients have been available. Therefore, we followed recommendations used in sports medicine [33].

The vibration ergometer consists of a conventional racing bicycle which is tightly mounted on a heavy platform. The bottom bracket is completely decoupled from the crank of the bicycle and firmly fixed to a separate plate which could be vibrated by a strong eccentric motor (see Figure 2; the blue arrow indicates where the plate with the bottom bracket is located). The bottom bracket and rear wheel are connected with a conventional bicycle chain (see Figure 2). Brackets are connected with the rear wheel so that a certain load could be added.

Without vibration, the device is an ergometer. The load was set to a very low value of 20 Watts. Since VET does not induce resonance phenomena of trunk and head [37], the frequency of vibration was reduced to 15 Hz. The vibration amplitude did not exceed 4 mm.

A complete vibration ergometry training session (VET) lasted 12 min and had three parts: a first vibration session of 5 min, followed by a pause of 2 min, and a further vibration session of 5 min. During the entire VET, patients had to sit on the device. For safety reasons, patients wore a climbing belt which was linked to a thick rod firmly attached to the ceiling of the laboratory. Even when the patient would slip out of the pedals, he/she could not fall from the ergometer.

### 2.4. Botulinum Toxin Injections

For both patient groups (VG and CG group), the dose units of the three different BoNT/A preparations (abobotulinum neurotoxin type A (aboBoNT/A), Ipsen^®^, Paris, France; onabotulinum neurotoxin type A (onaBoNT/A), Allergan^®^, Irvine, CA, USA; incobotulinum neurotoxin type A (incoBoNT/A), Merz Pharmaceuticals^®^, Frankfurt am Main, Germany)) used for treatment of legs and/or arms are presented in Table 1. For comparison purposes, doses were transformed into unified dose units (uDU) by dividing the aboBoNT/A doses by 3 and leaving the inco- and onaBoNT/A doses unchanged, following consensus recommendations [38].

### 2.5. Parameters of Gait Analysis

Gait was analyzed by means of the Ultraflex^®^ (UFG^®^) gait analysis system (Infotronic^®^, NL-7650 AB Tubbergen, The Netherlands) which consists of soft tissue shoes with a solid, but flexible, plate containing eight force transducers which quantitatively measure vertical ground reaction forces (GRF curves) and temporal patterns of foot–ground contact. These shoes are strapped over the feet or shoes of the patients. Thin cables connect the UFG^®^ shoes to a lightweight microprocessor which was tightly attached to a belt strapped around the belly of the patient. Patients were allowed to perform a short walk of less than 8 m to become familiar with the UFG^®^. During VET, the UFG^®^ shoes were taken off.

Patients had to walk a distance of 20 m, starting from a standing position with feet placed together. For data analysis, the first two steps and the last two steps were eliminated.

Eleven gait parameters were chosen for further analysis: in addition to gait velocity (Unit: meter (m)/second (s); VEL) and cadence (unit: 1/s; CAD), single support (unit: s; SS), double support (unit: s; DS), stance time (unit: s; STAN), and step time (unit: s; STEP) of the more (A) and the less-affected (NA) leg (SSAL, SSNAL; DSAL, DSNAL; STANAL, STANNAL; STEPAL, STEPNAL) were extracted from the long list of automatically delivered parameters of the UFG^®^. As the eleventh parameter, double stride length (unit: m; STRIDEL) was calculated from the number of steps and the distance.

### 2.6. Statistics

Differences in the distribution of sex and the side of the more affected leg were tested between the VG and CG group by means of the chi-square test. A U test was used to compare age, mean dose of leg and mean dose of arm treatment in VG and CG.

The design of the study allowed for the comparison of the efficacy of a single BoNT/A injection in the mildly affected VG and the more severely affected CG patients (Section 3.2). The effect of a single VET session as well as two VET sessions could be analyzed in the VG patients (Section 3.4 and Section 3.5). Since VG patients were analyzed with vibration and 6 months later without vibration, the efficacy of the combination of VET and BoNT/A injection could be compared to the effect of a single BoNT/A injection without vibration (Section 3.6).

A two-group repeated measurement (RM)-ANOVA was calculated to compare walking parameters according to the control protocol of VG and CG patients. A single group RM-ANOVA was used to analyze walking performance according to the vibration protocol in VG. All tests were part of the commercially available SPSS^®^ software package (version 25, IBM^®^, Armonk, NY, USA).

## 3. Results

### 3.1. Comparison of the Vibration Group (VG) and the Control Group (CG)

In the VG group, the mean age was 45 years. Three patients were males, four were females. Four patients were mainly affected on the right side of the body, three patients mainly on the left side. The mean dose used for leg muscle injection was 158 uDU, and the mean dose for arm muscle injections was 80 uDU.

In the CG group, the mean age was 58 years. Three patients were males, four were females. Four patients were mainly affected on the right side of the body, three patients mainly on the left side. The mean dose used for leg muscle injection was 177 uDU, and the mean dose for arm muscle injections was 126 uDU.

There was a tendency towards a younger age in the VG group, but no significant difference was found (*p* = 0.07). No significant differences were found for the sex distribution or the distribution of the more affected side of the body. Unified dose units used for the treatment of leg or arm spasticity did not differ between both groups (*p* = 0.668 for the leg and *p* = 0.509 for the arm; compare Table 1).

### 3.2. Comparison of VG and CG When Analyzed According to the Control Protocol

When walking according to the control protocol (W1, W3) was compared between the VG and CG group (mean values and standard deviations are presented in Table 2), VG patients (full symbols in Figure 2) walked with a significantly (*p* < 0.001) faster gait speed (Figure 2, left upper part), had a significantly higher (*p* < 0.05) cadence (Figure 2, right upper part), walked with a significantly (*p* < 0.01) longer stride length (Figure 2, left lower part), and had a significantly (*p* < 0.05) shorter double support (Figure 2, right lower part) compared to the CG patients (open symbols).

Between-group contrasts were significant (*p* < 0.05) for gait velocity and stride length when walking before (W1) and after (W3) the BoNT/A injections were compared. Four weeks after BoNT/A, VG patients (full circles) walked significantly faster with longer stride lengths, whereas CG patients walked with a non-significant reduction in gait speed without changes in stride length (open circles) (Figure 3).

### 3.3. Walking According to the Control and Vibration Protocol in a Single VG-Subject

In Figure 4, data of a single subject (V1 of VG) are presented. This 71-year-old male patient was affected on the left side and walked without a cane. The peak GRF curves on the left (L) affected side were much lower than on the less-affected right (R) side. Six different walking conditions are presented from top to bottom: the three walking conditions without vibration, NV-W1, NV-W2, and NV-W3 (left side), and the three walking conditions with vibration, V-W1, V-W3, and V-W4 (right side). For each walking condition, a pair of time segments of original recordings of the GRF curves (left leg (L), upper curve of the pair); right leg (R), lower curve of the pairs) (columns 1 and 3) and the corresponding overlays of all GRF curves of single steps (normalized to step duration) of a recording (columns 2 and 4) are presented. On the ordinate, always the ground reaction force in Newtons (N) is presented. For the sake of comparison, the 500 N level is presented as a horizontal bar for the overlays of the GRF curves of the affected leg.

Walking (without vibration) before (W1) and after (W2) a BoNT/A injection, with 800 U aboBoNT/A into the leg and 200 U aboBoNT/A into the arm muscles, and about 4 weeks later (W3) (compare columns 1 and 2) did not lead to clinically relevant changes in walking. The peak forces of the affected side remained similar for W1, W2 and W3 (compare the horizontal bars in column 2).

However, when the patient walked according to the vibration protocol (compare columns 3 and 4 in Figure 4) gait speed and therewith GRF considerably increased after 2 VET sessions (compare V-W1 with V-W2 and the horizontal bars on the right side). After BoNT/A-injection 4 weeks later, peak GRF on the affected side further increased (Figure 4, V-W3). The modulation of the GRF curves was also improved: the two physiological peaks of the GRF curves of the affected leg became more clearly distinguishable in W4 (compare V-W1 with V-W4 in Figure 4). This patient was one of four VG patients who reported an additional improvement with the combined application of VET and BoNT/A.

### 3.4. Comparison of Walking Before and After a Single Vibration Session

When walking was analyzed according to the vibration protocol gait parameters, it could be compared before and after a single session of VET twice: V-W1 and V-W2 were compared before and after VET1, as well as V-W4 with V-W5 before and after VET3 (see Figure 1). For all 11 parameters no within-group contrast was found for these two comparisons, V-W1/V-W2 and V-W4/V-W5 (see mean values in Table 3).

### 3.5. Comparison of Walking Before and After Two Vibration Sessions

When walking before (V-W1) and after (V-W3) the two sessions of vibration (VET1 and VET2) were compared, a significant within-group contrast was observed for gait speed (Figure 5, upper left) and cadence (Figure 5, upper right). Two sessions of VET resulted in a significant increase in gait speed (*p* < 0.023) and cadence (*p* < 0.043), as indicated by stars in Fig. 5. While double support showed a continuous but non-significant decrease with vibration (Figure 5, lower right), stride length did not exhibit any relevant changes (Figure 5, lower left).

### 3.6. No Additional Effect of the Combination of BoNT/A-Injections and Vibration

When the interaction between the effect of BoNT/A-injection therapy and VET was analyzed in the VG group, comparisons were made between baseline walking (V-W1) and walking four weeks later (V-W4), following VET both before (VET1) and after (VET2) the BoNT/A-injection on the one hand (e.g., change in gait velocity = 0.711–0.696). These were compared to the changes observed between baseline walking (NV-W1) and walking four weeks after BoNT/A-injection without vibration (NV-W3) on the other hand (e.g., change in gait velocity = 0.727–0.623). A positive change (improvement) was observed in both comparisons. However, no significant differences were identified in any of the 11 gait parameters analyzed between the vibration and the non-vibration conditions (for details, see the upper part of Table 2 and Table 3).

## 4. Discussion

### 4.1. General Remarks on the Analysis of Walking in Hemiparetic Patients

As can be seen from the GRF curves presented in Figure 4, patients with hemiparesis tend to shift their center of gravity to the less-affected side [39] but compensate for this shift of the lower body by bending the upper trunk to the more affected side [40]. Injections of botulinum toxin into the affected arm reduce the trunk lateral bending or “pushing of the upper trunk” to the more affected side and therewith improve gait velocity [41]. Therefore, BoNT-injection therapy of the arm has to be controlled when improvement of gait is analyzed. In the present study, no significant difference of mean unified doses of BoNT/A used for arm or leg treatment was found between the VG and CG group (see Table 1). In a previous comparison of the combined use of vibration and botulinum toxin in patients with multiple sclerosis, no details of BoNT/A treatment were given either for legs or for arms [25].

### 4.2. Previous Studies on the Combined Use of Botulinum Toxin and Vibration

The application of vibration has been combined with botulinum toxin for different indications. In cosmetic therapy, vibration has been used to reduce pain during BoNT injections [42]. Vibration has been tested to see whether it can compensate for the bone loss resulting from muscle disuse after BoNT injections in mice [30,31]. BoNT injections have also been combined with vibration to increase our knowledge on the tonic vibration reflex in writer’s cramp [43] or the vibration-induced facilitation of motor evoked potentials in spasmodic torticollis [44]. Some knowledge about the increase of the efficacy of BoNT/A injections by 14 days of conditioning VET came from a detailed investigation of a single normal subject [45]. There is one more study analyzing the combined use of botulinum neurotoxin and WBV on 60 spastic diplegic children [46] (for details see Section 4.5).

### 4.3. BoNT/A-Injections Improved Gait in the Less-Affected VG-Patients

It has been recommended by the AAN as well as the RCP to use BoNT/A injections for effective reduction of muscle tone in patients with spasticity [4,5]. However, reduction of muscle tone does not imply that a functional benefit results from BoNT treatment of lower leg spasticity. In general, it is difficult to demonstrate an increase of gait velocity by a therapeutic intervention in post-stroke spasticity [6]. Although the number of patients in VG was small (n = 7), a significant increase of mean gait velocity of 0.054 m/s from NV-W1 (0.673 m/s) to NV-W3 (0.727 m/s) was found (see Table 2). This is higher than 0.040 m/s, above which a clinically relevant improvement of gait velocity is noticed by the patient [6].

However, in the more affected, significantly slower walking CG patients (NV-W1: 0.403 m/s) no significant increase of gait velocity was found after BoNT/A injections (see Figure 3 and Table 2), although no difference in demographical as well as treatment-related data could be detected between the VG and CG group (see Table 1). This suggests that less-affected hemiparetic patients have a better benefit from a single BoNT/A injection than the more affected ones. But this does not imply that severely affected patients do not have any functional benefit from BoNT/A injections. It was recently demonstrated that, in 24 BoNT/A-naïve, severely affected post-stroke patients, a significant increase of gait velocity from 0.29 m/s to 0.32 m/s could be achieved by injections of 800 U aboBoNT/A either into the soleus or the gastrocnemius muscle [47]. But, in contrast to these BoNT-naïve patients, the VG as well as the CG patients in the present study had already been treated with BoNT/A on a regular basis every 3 months, so that they appeared to be in a steady-state, with little modulation of the peak effect around week 4. This influence of the duration of pre-treatment on the outcome after a therapeutic intervention has to be taken into account in following studies.

### 4.4. Vibration Ergometry Improves Gait

Compared to other studies on the effect of vibration training on spasticity, the intensity of vibration in the present study is very low [19,21,22,23,24,25,46]. Chan et al. [21] used a 12 Hz stimulation, a displacement of 4 mm, and a duration twice as long (10 min vibration, followed by a pause of 1 min, followed by a second period of 10 min of vibration) as the present VET. Therefore, it is not surprising that no significant effect was observed after a single vibration session (VET1 and VET3) in the present study.

However, after two sessions of vibration (VET1 and VET2), even after BoNT-injection which may have been slightly painful and thereby may have enhanced spasticity, a significant improvement of gait speed was observed (Figure 5). The double support (DS) of both legs was reduced. Duration of DS (in the present study) is the time to prepare shift of the body to the other side and to prepare the swing phase. Reduction of DS allows the patient to perform more steps per time unit, to increase cadence and therewith gait velocity.

This immediate effect of vibration has already been observed after a single session of whole-body vibration in post-stroke [21] and ICP patients [19]. Gait velocity, timed up and go test, as well as 10 m walk test were significantly improved, as well as uneven body weight posture [21]. The improvement of spasticity of ankle plantar flexors lasted up to 2 h after a single WBV session [19].

BoNT/A injections significantly improved stride length (Figure 3, left lower part) whereas vibration mainly influenced cadence (Figure 5, right upper part). The question therefore remains whether the combination of vibration and BoNT/A neutralizes each other or induces an additional effect.

### 4.5. The Combination of VET and BoNT/A-Injections Did Not Show an Additional Improvement

When walking was tested four weeks after BoNT/A injection in the VG group, gait velocity had significantly increased, regardless of whether patients underwent VET before and after BoNT/A injection (comparison between V-W1 and V-W4) or not (comparison between NV-W1 and NV-W3). There was no significant difference between these two conditions. Although two sessions of VET before and after BoNT/A injection had significantly increased gait velocity measured immediately after VET2, an additional enhancement of the efficacy of BoNT/A four weeks later could not be detected.

On the other hand, we have been able to demonstrate recently that daily VET over 10 days significantly enhanced the efficacy of incoBoNT/A injected into the extensor digitorum brevis (EDB) muscle of the foot [45]. We therefore conclude that the two sessions of VET were not intensive enough to influence the efficacy of BoNT/A uptake to such an extent that walking was significantly improved. This fits to the observation that a single session of VET did not have any influence on walking. Two sessions of VET had an immediate effect on walking, but did not have any long-term effect.

In a previous study, a significant influence of vibration on spasticity in multiple sclerosis patients was demonstrated [25]. However, also in this study, the combination of vibration and BoNT/A injections did not show as good a result as each of vibration or BoNT/A injections. However, the underlying reason is different compared to the present study. Patients who received BoNT/A injections as well as WBV had the lowest baseline scores, so the improvement was lower than in the other two subgroups. However, the relative improvement compared to baseline was the highest among the patients with WBV and BoNT/A application (for details see [25]). In the present study, the patients were their own controls, so a difference in baseline values cannot account for the missing significance.

In another study [46] on the combined use of vibration and botulinum toxin injection therapy, 60 spastic diplegic children aged between 2 and 5 years were equally split up into an experimental and a control group. Both groups received 5 daily courses of conventional training, 5 days per week for 3 weeks. The experimental group additionally received 2 min of WBV 3 to 4 times per day, 5 days per week, for 5 weeks. Training started the next day after BoNT injection. For assessment, several parameters were used, including the modified Tardieu scale (MTS) and dimensions D and E of the gross motor function measurement (GMFM-88). Children were assessed 1, 3, and 6 months after BoNT injection. In both groups a significant improvement of all parameters was observed, which was more pronounced in the experimental group, with the only exception being that the GMFM-88 did not show a significant difference between both groups 1 month after the BoNT injection [46]. This reduction of the effect size, at the time when the desensitizing effect of BoNT on muscle spindles is maximal, fits with the results of the present study.

We therefore believe that, in the present study, vibration was not applied for a sufficient duration or intensity to adequately prepare motor unit endplates for improved BoNT/A uptake. Taking previous recommendations [33] into account, it appears that we were overly cautious and selected vibration parameters that were too low.

## 5. Conclusions

Both vibration ergometry training as well as BoNT/A-injections improved gait in patients with asymmetric lower leg spasticity. Functional improvement in spasticity was observed in mildly affected patients but not in those with moderate spasticity. A pre- and post-BoNT/A-injection vibration session lasting 10 min proved insufficient to enhance the efficacy of BoNT/A injections in patients undergoing continuous BoNT injection therapy every three months.

## 6. Strengths and Limitations of the Study

This study reveals a short-term functional benefit of vibration and a long-term functional benefit of BoNT/A-injections in mildly affected patients with asymmetric leg spasticity. The difference in response to BoNT/A-application in mildly and moderately affected patients offers an explanation as to why it is difficult, in general, to demonstrate a functional benefit after BoNT/A injections.

Compared to other studies on the effect of vibration on lower leg spasticity, the intensity of vibration was low in the present study. Furthermore, a time span of at least 6 months between the analysis according to the vibration protocol and the analysis according to the control protocol might have had a relevant influence. We therefore recommend performing a further study in a cross-over design with a more intensive pre-conditioning vibration stimulation over days to demonstrate enhancement of BoNT/A efficacy by vibration.

## Figures and Tables

**Figure 1 jfmk-10-00041-f001:**
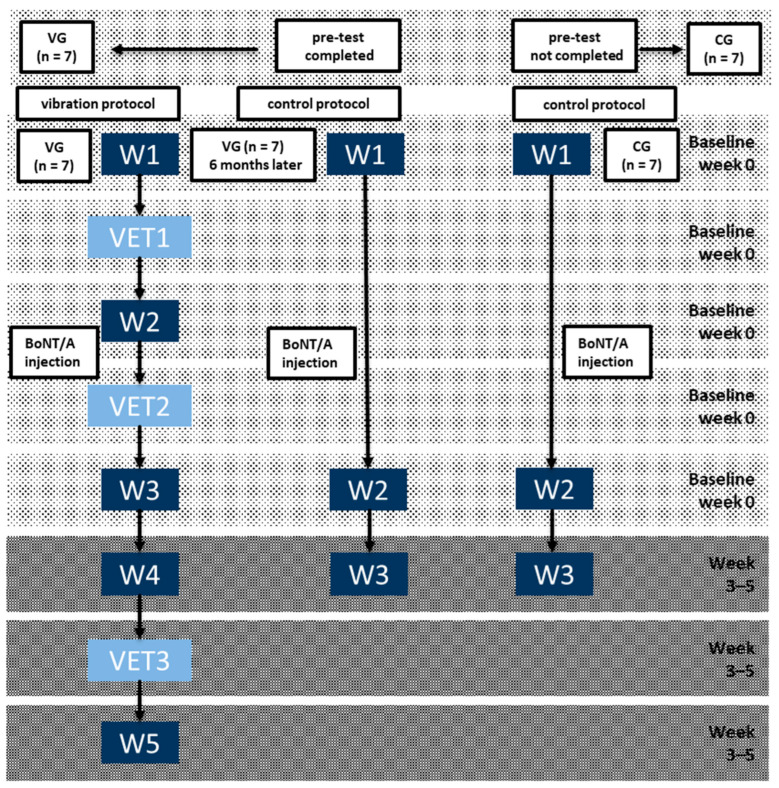
Design of the study. In the first step, pre-test VET was used to separate patients into those who managed to complete the pre-test (VG group; **left side**) and those who did not complete the pre-test (CG-group; **right side**). VG patients were first analyzed according to the vibration protocol and, 6 months later, also according to the control protocol. CG-patients were analyzed according to the control protocol only. When patients were investigated according to the vibration protocol (**left side**), they walked a first time at the baseline visit (W1), received the first VET1, walked a second time (W2), received their routine BoNT/A injection, received a second VET2, and walked a third time (W3). About 3–5 weeks later, they walked again (W4), received a third VET3, and walked a fifth time (W5). When patients were analyzed according to the control protocol (**middle part** and **right side**), patients had to walk (W1), receive their routine BoNT/A injection, and had to walk a second time (W2). About 3–5 weeks later, they had to walk a third time (W3).

**Figure 2 jfmk-10-00041-f002:**
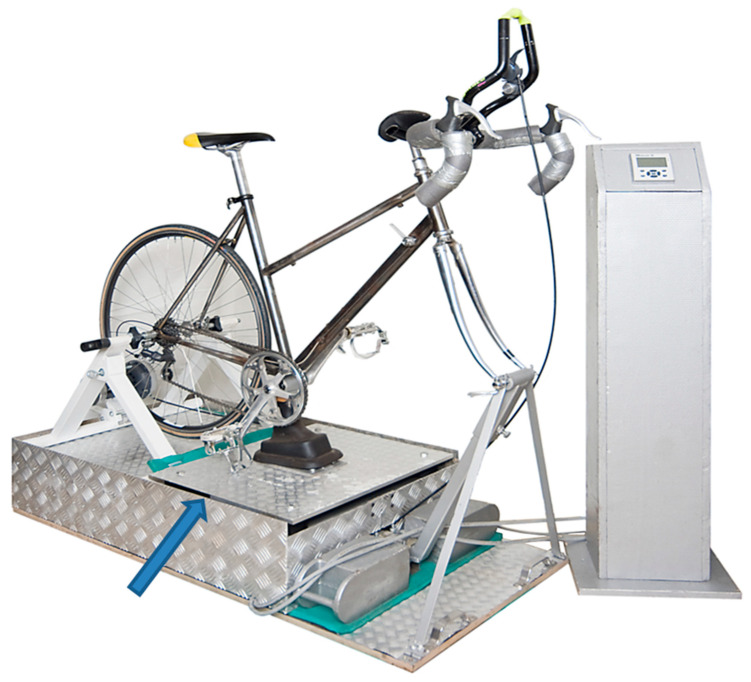
Photo of the vibration ergometer device used in the present study. A conventional racing bicycle was tightly mounted on a heavy platform with its bottom bracket and pedals being completely decoupled from the crank. The bottom bracket was firmly fixed to a separate plate which could be vibrated by a strong eccentric motor (blue arrow indicates where the plate with the bottom bracket is located). The bottom bracket and rear wheel were connected with a conventional bicycle chain.

**Figure 3 jfmk-10-00041-f003:**
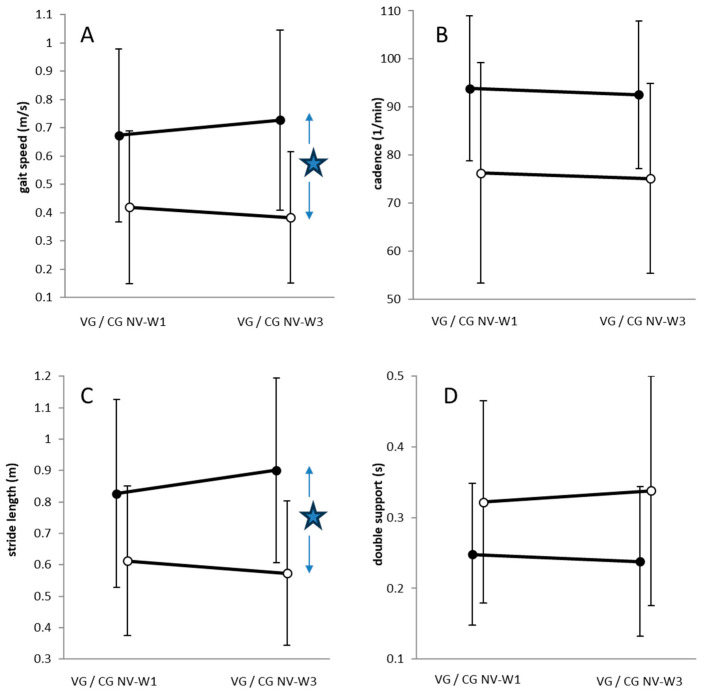
Comparison of mean values and standard deviations (error bars) of 4 gait parameters (gait velocity (**A**), cadence (**B**), stride length (**C**), double support of the affected leg (**D**) of walking W1 and walking W3 in VG patients (full circles) and CG patients (open circles) when patients were analyzed according to the control protocol. Apart from the obvious between group difference, a significantly (*p* < 0.05) higher change in gait speed (**A**) and stride length (**C**) of walking W3 compared to walking W1 was observed in the VG patients compared to the CG patients. This is indicated by the two small arrows and the star in A and C.

**Figure 4 jfmk-10-00041-f004:**
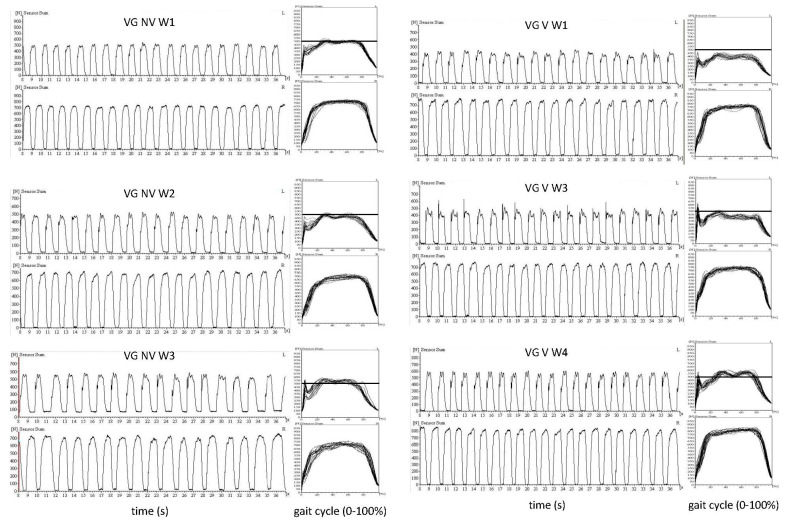
This figure shows ground reaction force (GRF) curves for patient V1, a left-sided hemiparetic patient, under two protocols: the control protocol (**left side**) and the vibration protocol (**right side**). **On the left side (Control Protocol),** GRF curves were recorded during three walking tests: baseline walk (**NV-W1**, **upper part**), immediately after BoNT/A injection (**NV-W2**, **middle part**), and four weeks post-injection (**NV-W3**, **lower part**). The second column displays the overlaid GRF curves of single steps after normalization of step duration comparison. The results show no clear differences across the three conditions: the peak ground reaction forces on the affected (left) side remained relatively unchanged, as highlighted by the horizontal bars. These findings indicate no notable improvement in walking performance under the control protocol. **On the right side (Vibration Protocol),** GRF curves were recorded during three walking tests: baseline walk (**V-W1**, **upper part**), immediately after two vibration ergometry training (VET) sessions and BoNT/A injection (**V-W3**, **middle part**), and four weeks post-intervention (**V-W4**, **lower part**). In the fourth column, the corresponding normalized GRF curves are overlaid for comparison. Following two VET sessions, the peak ground reaction forces on the affected side increased noticeably (V-W3). By four weeks post-intervention (V-W4), the peak forces further improved and the GRF modulation (two distinct peaks) becomes clearer, indicating better weight-shifting and gait dynamics. In summary, while no changes occurred under the control protocol, the vibration protocol led to measurable functional improvements, including increased peak ground forces and enhanced GRF modulation, which were noticed by the patient, demonstrating the potential short- and long-term benefits of combining VET with BoNT/A injection.

**Figure 5 jfmk-10-00041-f005:**
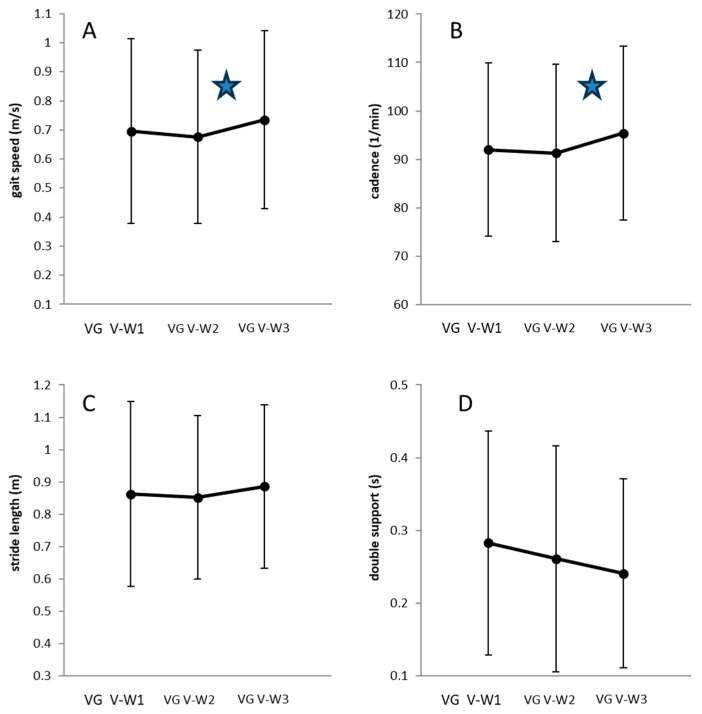
Comparison of mean values and standard deviations (error bars) of 4 gait parameters (gait velocity (**A**), cadence (**B**), stride length (**C**), double support of the affected leg (**D**) of walking V-W1, walking V-W2 and walking V-W3 in VG patients when walking was analyzed according to the vibration protocol. Gait speed (**A**) and cadence (**B**) significantly (*p* < 0.05) increased after the second VET as indicated by the stars in (**A**,**B**).

**Table 1 jfmk-10-00041-t001:** Demographical and treatment-related data of the VG group (patients V1…V7) and the CG group (patients C1…C7).

Patient	Age (yrs)	Sex	CNS Lesion	Walking Aid	Affected Side	Dose of BoNT/A Leg Muscles	Dose of BoNT/A Arm Muscles
V 1	71	m	R THL HR	none	L HP	800U aboBoNT	200U aboBoNT
V 2	49	f	L BG HR	none	R HP	90U onaBoNT	110U onaBoNT
V 3	49	f	L HS IS	CANE	R HP	400U aboBoNT	600U aboBoNT
V 4	32	m	ICP	none	R HP	300U aboBoNT	0
V 5	41	f	R CAD IS	none	L HP	600U aboBoNT	400U aboBoNT
V 6	48	m	R MI IS	none	L HP	215U incoBoNT	185U incoBoNT
V 7	25	f	L fTBI	none	R HP	300U aboBoNT	0
C 1	57	f	R fTBI	SOP	L HP	500 U aboBoNT	0
C 2	62	f	R HS HR	AFO, CANE	L HP	500 U aboBoNT	0
C 3	66	m	L HS IS	none	R HP	1000U aboBoNT	0
C 4	46	f	L HS HR	AFO, SOP	R HP	150 U onaBoNT	150 U onaBoNT
C 5	45	m	L MI IS	none	R HP	750 U aboBoNT	800 U aboBoNT
C 6	60	m	L HS IS	none	R HP	300 U aboBoNT	700 U aboBoNT
C 7	70	f	R HS IS	none	L HP	70 U incoBoNT	230 U incoBoNT

CNS = central nervous system; f/m = female/male; R = right; L = left; fTBI = focal traumatic brain injury; THL = thalamic; BG = basal ganglia; HS = hemisphere; ICP = infantile cerebral palsy; CAD = dissection of the carotid artery; MI = infarct of the medial cerebral artery; IS = ischemic stroke; HP = hemiparesis; U = mouse units; HR = hemorrhage; SOP = support by another person (by holding one hand to improve safety); AFO = ankle-foot orthosis.

**Table 2 jfmk-10-00041-t002:** Walking according to the control protocol in VG and CG.

VG	NV-W1 (MV/SD)		NV-W2 (MV/SD)		NV-W3 (MV/SD)	
VEL (m/s)	0.673	0.305	0.711	0.310	0.727	0.318
CAD (/min)	93.86	15.09	94.14	14.69	92.57	15.33
STRIDEL (m)	0.826	0.299	0.873	0.296	0.901	0.294
SSAL (s)	0.345	0.036	0.321	0.058	0.359	0.041
SSNAL (s)	0.500	0.117	0.464	0.062	0.497	0.094
DSAL (s)	0.248	0.100	0.270	0.108	0.238	0.106
DSNAL (s)	0.220	0.052	0.251	0.088	0.240	0.101
STANAL (s)	0.814	0.136	0.842	0.176	0.837	0.196
STANNAL (s)	0.968	0.247	0.985	0.231	0.974	0.254
STEPAL (s)	0.594	0.105	0.591	0.106	0.598	0.100
STEPNAL (s)	0.720	0.156	0.715	0.149	0.736	0.175
**CG**	**NV-W1 (MV/SD)**		**NV-W2 (MV/SD)**		**NV-W3 (MV/SD)**	
VEL (m/s)	0.419	0.270	0.415	0.244	0.383	0.233
CAD (/min)	76.29	22.98	75.86	23.36	75.14	19.75
STRIDEL (m)	0.612	0.238	0.616	0.207	0.573	0.230
SSAL (s)	0.339	0.086	0.361	0.121	0.336	0.108
SSNAL (s)	0.503	0.096	0.515	0.179	0.522	0.093
DSAL (s)	0.322	0.143	0.308	0.122	0.338	0.162
DSNAL (s)	0.525	0.338	0.513	0.354	0.486	0.383
STANAL (s)	1.187	0.422	1.182	0.419	1.160	0.452
STANNAL (s)	1.350	0.469	1.336	0.465	1.346	0.456
STEPAL (s)	0.661	0.179	0.670	0.176	0.675	0.172
STEPNAL (s)	1.028	0.357	1.028	0.367	1.008	0.336

MV = mean value, SD = standard deviation, m = meter, min = minute, s = second, VEL = gait velocity, CAD = cadence, STRIDEL = stride length (time for 1 step of the left and 1 step of the right leg), SSAL = single support of the affected leg, SSNAL = single support of the less-affected leg, DSAL = double support of the affected leg, DSNAL = double support of the less-affected leg, STANAL = stance time on the affected leg, STANNAL = stance time on the less-affected leg, STEPAL = step time of the affected leg, STEPNAL = step time of the less-affected leg.

**Table 3 jfmk-10-00041-t003:** Walking according to the vibration protocol in VG.

VG	V-W1 (MV/SD)		V-W2 (MV/SD)		V-W3 (MV/SD)		V-W4 (MV/SD)		V-W5 (MV/SD)	
VEL (m/s)	0.696	0.318	0.676	0.298	0.735	0.306	0.711	0.323	0.699	0.315
CAD (/min)	92.00	17.90	91.29	18.30	95.43	17.97	93.00	17.00	92.43	17.85
STRIDEL (m)	0.862	0.286	0.852	0.253	0.886	0.253	0.872	0.297	0.867	0.269
SSAL (s)	0.342	0.058	0.357	0.070	0.346	0.059	0.347	0.033	0.349	0.038
SSNAL (s)	0.498	0.116	0.524	0.120	0.506	0.139	0.493	0.107	0.512	0.131
DSAL (s)	0.283	0.154	0.261	0.155	0.241	0.130	0.261	0.162	0.262	0.156
DSNAL (s)	0.232	0.072	0.227	0.057	0.207	0.036	0.236	0.067	0.226	0.058
STANAL (s)	0.859	0.203	0.845	0.196	0.794	0.155	0.844	0.206	0.837	0.182
STANNAL (s)	1.013	0.293	1.012	0.300	0.954	0.278	0.989	0.309	1.001	0.310
STEPAL (s)	0.626	0.164	0.618	0.154	0.588	0.130	0.608	0.160	0.611	0.150
STEPNAL (s)	0.730	0.163	0.751	0.171	0.713	0.167	0.728	0.161	0.738	0.171

MV = mean value, SD = standard deviation, m = meter, min = minute, s = second, VEL = gait velocity, CAD = cadence, STRIDEL = stride length (time for 1 step of the left and 1 step of the right leg), SSAL = single support of the affected leg, SSNAL = single support of the less-affected leg, DSAL = double support of the affected leg, DSNAL = double support of the less-affected leg, STANAL = stance time on the affected leg, STANNAL = stance time on the less-affected leg, STEPAL = step time of the affected leg, STEPNAL = step time of the less-affected leg.

## Data Availability

Data available on request due to restrictions privacy or ethical. The data are part of the doctoral thesis of JB and will be available after the official end of her dissertation procedure and on request from the corresponding author.

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
