# Peer review of "Effects of Combined Vibration Ergometry and Botulinum Toxin on Gait Improvement in Asymmetric Lower Limb Spasticity: A Pilot Study"

_jfmk, 2025, doi:10.3390/jfmk10010041_

Round 1
Reviewer 1 Report
Comments and Suggestions for Authors
Thank you for the opportunity to review this manuscript - the topic is the effects of combined vibration ergometry and botulinum toxin on gait improvement. I have some concerns about this manuscript. Please refer to the following points for revision.
More specific feedback is as follows:
Methods
Line-162:「Compared to other studies on segmental or whole-body vibration [21] vibration parameters used in the present study were low (see discussion).」
I think this sentence should be written in the discussion, not in the method.
Line-189:「Eleven gait parameters were chosen for further analysis: in addition to gait velocity (m/s; VEL) and cadence (1/s; CAD), single support (s; SS), double support (s; DS), stance time (s; STAN) and step time (s; STEP)」
Is the ‘s’ an abbreviation for second? Abbreviations should be spelt out for the first time. In addition, it would be better if the units were stated in a table.
Results
Line-213:「There was a clear tendency for a younger age in the VG-group, but no significant difference (p=.07)」
It is not statistically significant and the number of patients is so small. I think it is not appropriate to describe this as a ‘clear tendency’. Also, I think the P-value should be stated as ‘p = 0.07’.
Table
Table 1 is difficult to read and difficult to compare between groups. It is better not to draw a line between MV and SD, but to separate them with brackets. Instead of creating a table for each CG and VG, it would be better to create a column for CG and VG so that they can be compared side by side.
Figure
The explanation in Figure 1 is understandable, but the background colour and font size give the impression of being difficult to read. If you agree with this opinion, please revise it.
Other
Hyphenations are inserted in some unnecessary places. Please check them.
Author Response
Thank you for the opportunity to review this manuscript - the topic is the effects of combined vibration ergometry and botulinum toxin on gait improvement. I have some concerns about this manuscript. Please refer to the following points for revision. More specific feedback is as follows: Methods Line-162:「Compared to other studies on segmental or whole-body vibration [21] vibration parameters used in the present study were low (see discussion). I think this sentence should be written in the discussion, not in the method. Line-189:「Eleven gait parameters were chosen for further analysis: in addition to gait velocity (m/s; VEL) and cadence (1/s; CAD), single support (s; SS), double support (s; DS), stance time (s; STAN) and step time (s; STEP)」 Is the ‘s’ an abbreviation for second? Abbreviations should be spelt out for the first time. In addition, it would be better if the units were stated in a table. Results Line-213:「There was a clear tendency for a younger age in the VG-group, but no significant difference (p=.07)」 It is not statistically significant and the number of patients is so small. I think it is not appropriate to describe this as a ‘clear tendency’. Also, I think the P-value should be stated as ‘p = 0.07’. Table Table 1 is difficult to read and difficult to compare between groups. It is better not to draw a line between MV and SD, but to separate them with brackets. Instead of creating a table for each CG and VG, it would be better to create a column for CG and VG so that they can be compared side by side. Figure The explanation in Figure 1 is understandable, but the background colour and font size give the impression of being difficult to read. If you agree with this opinion, please revise it. Other Hyphenations are inserted in some unnecessary places. Please check them. |
We agree with reviewer 1:
This sentence is transferred to the discussion.
We again agree with reviewer 1:
The use of the single “s” without previous explanation is misleading. “s” is now explained in the list of abbreviations. It is also explained in the text when it is used the first time. We have preferred to add “units:” for each parameter instead of adding another table. We hope that reviewer 1 is satisfied with that.
We again agree with reviewer1:
The “clear” is omitted now.
Table 1 is modified according to reviewer 1´s recommendations:
We have combined Table 1 A and 1B , have combined VG and CG in one column so that a better comparison is possible.
Thanks to reviewer 1 we have modified Fig. 1 so that the difference in the flow of investigations between VG and CG becomes more obvious.
Reviewer 1 is absolutely right. Especially in the list of abbreviations hyphenations had been inserted.
The authors are thankful to reviewer 1 for detailed reading and helpful comments |
Reviewer 2 Report
Comments and Suggestions for Authors
Aim of this study was to study combined effect of Botulinum neurotoxin type A (BoNT/A) injections and the new vibration ergometry training (VET) on improving functional mobility in patients with asymmetric lower limb spasticity.
Although authors described study protocol and results in a precise, scientific and methodical way, manuscript has important negative aspects that prevent its publication in this journal:
1) the sample of the two groups (7 patients each) is too small to be able to obtain scientifically valid conclusions, considering that the number of patients with Asymmetric Lower Limb Spasticity is very large (it's no rare syndrome)
2) the background and conclusions are based on a bibliography that is too old (only 4/5 out of 46 are from 5 years ago).
I believe that the work should be thoroughly revised before it can be published in this journal.
Comments on the Quality of English LanguageMinor revision
Author Response
Aim of this study was to study combined effect of Botulinum neurotoxin type A (BoNT/A) injections and the new vibration ergometry training (VET) on improving functional mobility in patients with asymmetric lower limb spasticity. Although authors described study protocol and results in a precise, scientific and methodical way, manuscript has important negative aspects that prevent its publication in this journal: 1) the sample of the two groups (7 patients each) is too small to be able to obtain scientifically valid conclusions, considering that the number of patients with Asymmetric Lower Limb Spasticity is very large (it's no rare syndrome) 2) the background and conclusions are based on a bibliography that is too old (only 4/5 out of 46 are from 5 years ago).
I believe that the work should be thoroughly revised before it can be published in this journal. |
In a statistical test even small numbers of patients can be used if the effect size is strong enough.
This is supported by the next point. Under the heading “botulinum toxin” or “botulinum neurotoxin” and “vibration” we have detected only 1 additional relevant publication by REN et al. (2019).
In this study 60 children with spastic diplegia were equally divided into a control and an experimental group. Children in the experimental group received intensive whole body vibration for 3 weeks in addition to convential training. Nevertheless, no difference was found between the control and the experimental group 1 month after baseline investigation, but significant differences emerged 2 and 5 months later. This underlines that the effect size of vibration and BoNT combination is small 4 weeks after a BoNT-injection when botulinum toxin is mostly effective.
This is a new aspect and pick-up in the discussion.
The authors are thankful to reviewer 2 that this aspect could be added and the discussion thoroughly revised. |
Round 2
Reviewer 1 Report
Comments and Suggestions for Authors
I think the authors have made sufficient improvements to the publication in response to my comments.
Reviewer 2 Report
Comments and Suggestions for Authors
New manuscript version satisfies my previous doubts; authors answered my previous comments precisely and scientifically.